# Environmental and Occupational Short-Term Exposure to Airborne Particles and FEV_1_ and FVC in Healthy Adults: A Systematic Review and Meta-Analysis

**DOI:** 10.3390/ijerph182010571

**Published:** 2021-10-09

**Authors:** Alan da Silveira Fleck, Margaux L. Sadoine, Stéphane Buteau, Eva Suarthana, Maximilien Debia, Audrey Smargiassi

**Affiliations:** 1Department of Environmental and Occupational Health, School of Public Health, University of Montreal, 2375 Chem. de la Côte-Sainte-Catherine, Montreal, QC H3T 1A8, Canada; alan.da.silveira.fleck@umontreal.ca (A.d.S.F.); margaux.sadoine@umontreal.ca (M.L.S.); maximilien.debia@umontreal.ca (M.D.); 2Centre for Public Health Research (CReSP), 7101 Av du Parc, Montreal, QC H3N 1X9, Canada; 3Institut National de Sante Publique du Québec (INSPQ), 190 Boul Crémazie E, Montreal, QC H2P 1E2, Canada; Stephane.Buteau@inspq.qc.ca; 4Research Institute of the McGill University Health Center, 2155 Rue Guy, Montreal, QC H3H 2L9, Canada; eva.suarthana@gmail.com; 5Centre de Recherche de l’Hôpital du Sacré-Coeur de Montréal (CRHSCM), 5400 Boul Gouin O, Montreal, QC H4J 1C5, Canada

**Keywords:** particulate matter, lung function, healthy adults, short-term, occupational exposures, environmental exposures

## Abstract

Background: No study has compared the respiratory effects of environmental and occupational particulate exposure in healthy adults. Methods: We estimated, by a systematic review and meta-analysis, the associations between short term exposures to fine particles (PM_2.5_ and PM_4_) and certain parameters of lung function (FEV_1_ and FVC) in healthy adults. Results: In total, 33 and 14 studies were included in the qualitative synthesis and meta-analyses, respectively. In environmental studies, a 10 µg/m^3^ increase in PM_2.5_ was associated with an FEV_1_ reduction of 7.63 mL (95% CI: −10.62 to −4.63 mL). In occupational studies, an increase of 10 µg/m^3^ in PM_4_ was associated with an FEV_1_ reduction of 0.87 mL (95% CI: −1.36 to −0.37 mL). Similar results were observed with FVC. Conclusions: Both occupational and environmental short-term exposures to fine particles are associated with reductions in FEV_1_ and FVC in healthy adults.

## 1. Introduction

Throughout the world, air pollution is a leading cause of mortality and morbidity [1], with particulate air pollution being responsible for around three million deaths each year [2]. This public health issue has emerged from the world’s progress and has been the subject of considerable attention for effects on health of both short-term and long-term exposure to air pollution [3].

Particulate matter (PM) is characterized by multiple components and size fractions (fine and coarse), of which distribution and proportion vary substantially depending on local emissions geography and meteorology [4,5]. This regional and temporal variability influences the magnitude of the health effects [6,7], as does the difference in the chemical composition of the particles and in the penetration into the respiratory tract [4,8,9]. Exposures to particulate matter have been linked to increased mortality, emergency room visits, and hospitalizations due to the exacerbation of cardio-respiratory diseases in children, the elderly, and adults [10,11,12,13]. PM acts on the development of noncommunicable lung disease [14], and it has been suggested to influence the susceptibility to acute lower respiratory tract infection [15]. The causal or likely causal relationships between long-term exposure to PM and all-cause and cardio-respiratory mortality have been well established by several organizations [1,16], including lung cancer by the International Agency for Research on Cancer [17]. Risks of asthma development have also been demonstrated [1,18].

Research on short-term exposures has shown associations with respiratory symptoms [19] and decreased lung function in individuals with pre-existing respiratory diseases [20,21]. Reduction in different lung parameters: FEV1 (−3.38 mL, 95% CI: −6.39 to −0.37) and PEF (−0.61 L/min, 95% CI: −1.20 to −0.01) has been reported in COPD patients with 10 µg/m^3^ increase in PM exposure [20]. However, the relationship between daily exposures to particles and lung function reductions is not established for healthy adults of the general and worker populations. Episodes of high environmental exposures to particles during short periods are ubiquitous and strongly related to society’s current urban organization model. Such exposures can occur while commuting [22], performing physical activity near a high traffic route [23,24,25], during episodes of high daily average concentrations of fine and ultrafine particles [26,27], and in microenvironments near transport hubs, roadways, underground train stations, and industrial sites [28,29,30,31]. In addition, millions of workers worldwide are daily exposed to processes and tasks associated with the emission of particles at concentrations higher than the typical urban background, such as welding fumes, forest fires, wood dust, and diesel engine exhaust [32,33,34,35], which are known to contribute to mortality by respiratory outcomes [36,37].

To our knowledge, no study has yet systematically reviewed the short-term effects of occupational exposures to airborne particles on FEV_1_ and FVC. Furthermore, there has been no attempt to compare lung function effects from environmental and occupational exposures to airborne particles, even though daily (24 h) and sub-daily (<24 h) exposures in these two contexts are shared by healthy adults. Such comparison may provide valuable insights into the relationship between short-term exposures to particles and lung function across different exposure ranges. In addition, of particular interest is how different the effects can be when considering the distinct sources, concentrations, and composition of particles across environmental and occupation settings. The objective of this study was to perform a systematic review and meta-analysis to estimate the associations between occupational and environmental short-term exposures to fine particles and changes in the lung function parameters most studied so far, specifically forced expiratory volume in one second (FEV_1_) and forced vital capacity (FVC), among healthy adults.

## 2. Materials and Methods

### 2.1. Registration

The protocol of this systematic review and meta-analysis was registered in PROSPERO (Registration Number: CRD42017078435). Moreover, the Preferred Reporting Items for Systematic Reviews and Meta-Analyses (PRISMA) checklist [38] was completed (Appendix A).

### 2.2. Search Strategy

The literature search included studies published in English between 1964 and 2020. The following electronic bibliographic databases were searched: Web of Science (Web of Science Core Collection and MEDLINE) and PubMed. Searches were last updated on 14 May 2020. In addition, we examined the reference lists of all included studies. The search included terms for the exposure to fine particles (i.e., respirable dust and PM2.5) and the selected outcomes lung function parameters: FEV1 and FVC. FEV1 refers to the quantity of air a person can exhale during the first second of a forced breath, while the FVC refers to the total amount of air exhaled during the spirometry test. These two indices were chosen given they are the most commonly investigated in studies associating air pollution exposures and lung function effects. We also added terms related to inflammation and exhaled nitric oxide fraction (FeNO), but no results related to the latter are reported as there were not enough occupational studies measuring FeNO to perform a meta-analysis or systematic review. Other terms were included to discard studies on animal models, children, in vitro models and long-term exposure studies. The complete search strategy and the keywords are presented in the Appendix A (Appendix A).

### 2.3. Inclusion and Exclusion Criteria

Studies were included if they repeatedly investigated acute respiratory effects (within 24 h after exposure) of short-term exposures (i.e., duration between 1 and 24 h) to fine particles in healthy adults of working age (i.e., between 18 and 60 years old). In terms of the study population, we restricted the review to healthy adults of working age to compare associations between occupational and environmental health studies. Studies with both healthy and non-healthy subjects were included if the authors mentioned that they controlled for health status or if results were reported by health status. We also considered in this review a few studies that included a small percentage of non-healthy subjects; these studies are well identified in the paper and were considered in the sensitivity analysis.

In terms of study design, we restricted the review to studies with repeated measurements of the outcomes because such design enabled separation of the effects of daily exposures from those of cumulative (long-term) exposures; therefore, cross-sectional studies were excluded. The selected study designs included panel and crossover environmental studies, as well as cross-shift occupational studies. Panel studies involve repeated measurements on each subject at specified short time intervals (i.e., daily); thus, each subject acts as his/her own control [20]. Crossover studies also involve repeated measurements, but the exposure situations are controlled by the researchers (e.g., cycling on high- and low- traffic routes) [25]. Occupational cross-shift studies involve the measurements of the outcome before and after the work.

We restricted the scope of this review to exposure to the mass concentration of PM2.5 (i.e., particles with a median aerodynamic diameter of 2.5 μm) and of respirable dust (i.e., PM4 particles with a median aerodynamic diameter of 4 μm), which are common classifications from the environmental and occupational studies, respectively. Although the size range of PM2.5 and PM4 is not exactly the same, both particulate matter fractions have a high capacity to penetrate deep into the alveolar region [39]. Furthermore, these different size fractions may not necessarily result significantly in different mass concentrations if the mean size distribution of the airborne particles is smaller than 2.5 µm.

Studies were excluded if: (1) they were based on reviews, they were experimental studies, case reports, letters, posters, and conference abstracts; (2) the study population was formed exclusively by children, the elderly or subjects with a pre-existing chronic respiratory disease such as asthma and Chronic Obstructive Pulmonary Disease (COPD); (3) the respiratory outcomes of interest were not measured within 24 hours after exposure; (4) exposure duration was not within 24 hours; (5) the studies reported only one measurement of the outcome per subject (i.e., no repeated measurements); (6) only size fractions other than PM2.5 and PM4 were measured; and (7) the exposure was focused on the measurement of environmental tobacco smoke.

### 2.4. Studies Selection and Data Extraction

The selection of the articles was performed in two rounds by two investigators (AF and MS). The first round consisted of a screening of all titles and abstracts. In the second round, the full texts of all potentially relevant studies were reviewed considering the inclusion and exclusion criteria. Potential divergences in the selection of the study were discussed and ultimately resolved by a third investigator (AS).

Data extraction was performed by one investigator (AF) and reviewed by a second one (MS). The following information was manually extracted: (1) authors and year; (2) study location; (3) study design (i.e., panel, crossover, and occupational cross-shift studies); (4) population (N, sex, % of smokers, and mean age); (5) exposure information such as type of measurement (i.e., personal/quasi-personal and central/near station), exposure duration, exposure context and type of particles; (6) physiological outcomes; (7) confounders and effect modifiers; and (8) results (mean concentration of particles and respiratory outcome results (e.g., estimate ± 95% CI or *t*-test result).

### 2.5. Quality Assessment

The assessment of the risk of bias was performed by two investigators (AF and MS) according to the Office of Health Assessment and Translation (OHAT) tool developed by the National Institutes of Environmental Health Sciences-National Toxicology Program [40]. Within each study, we evaluated the risk of bias across seven parameters divided as key criteria (i.e., exposure assessment, outcome assessment, and confounding bias, which is an item of the OHAT that includes the lack of consideration of important modifiers) and other criteria (i.e., selection bias, selective reporting, incomplete outcome data, and conflict of interest). The risk of bias for each parameter was evaluated as “low”, “medium”, “high”, or “not applicable”. The OHAT guideline recommends the exclusion of studies for which all the key criteria and most of the other criteria are characterized as “high”.

### 2.6. Meta-Analyses

Our initial goal was to perform a meta-analysis of results, notably to investigate whether associations between short-term exposure to fine particles and FEV1 and FVC differ between environmental and occupational settings. We thus considered separately environmental and occupational studies. However, the pooling of selected studies was limited by the different metrics used for the outcomes. Specifically, lung function parameters were expressed as: (a) absolute change (mL change); (b) percent change from a baseline or mean value (% change); (c) percent change of a log-transformed outcome (log % change); (d) percent change from a predicted value (%PV); and (e) percent change from a log-transformed predicted value (log %PV). The description of these five different outcome metrics identified in the studies are presented in the Appendix A (Appendix A). Pooling results of studies in a meta-analysis requires the measure of association to be expressed uniformly across studies [41]; therefore, these different metrics used for the lung function parameters could not be combined into a single meta-analysis of results.

We thus pooled studies separately according to the outcome metric. We computed meta-estimates when a minimum of five independent risk estimates was available. Due to the expected heterogeneity caused by the different study designs, populations and exposure characteristics, a random-effect meta-analysis was performed, thus assuming that the true effect size varies across studies. Meta-estimates, 95% confidence intervals and 95% prediction intervals were calculated for a 10 µg/m^3^ of particulate concentration; we used a similar increment given that we aimed at contrasting the effect across these two different settings. We assessed heterogeneity using the I2 statistic [42], whereas publication bias was examined using funnel plots [43]. A meta-regression of the covariables against the outcome was not performed as some parameters (such as age) were very similar between studies or even missing (smoking status), and sample size was too small.

In some occupational cross-shift studies, an estimate of association from a regression model was not reported [33,44,45,46,47,48]. Instead, the authors reported the difference between the mean outcome response postexposure and pre-exposure (with a t-test comparing both measurements). For these cases, we used the reported mean response and level of exposure to calculate the effect for a 10 µg/m^3^ increase in the pollutant concentration.

In some studies, measurement of the outcome was made at several time points after the exposure ended [24,25,29,30,49,50,51]. In this review, the main series of meta-analyses included estimates for the time point immediately after exposure ended, as this time point was the most frequently measured across studies. Results of other time periods, when available, are presented in the Appendix A (Appendix A). 

In the sensitivity analyses, we performed a leave-one-out test to explore the influence of each study included on the meta-estimate, and on the I2 statistic. Sensitivity analyses were also performed to explore the influence of the studies that had a small percentage of non-healthy subjects in its population. In addition, subgroup analyses were performed to explore the influence of key study criteria such as the type of measurement, duration of exposure and study design. The heterogeneity variance was assessed by the DerSimonian and Laird method. Statistical analyses were all performed using Review Manager 5.3 and the metafor package for R (version 3.4.1, R Foundation for Statistical Computing, Vienna, Austria) [52]. Results of studies that could not be included in the meta-analysis are described in the Appendix A (Appendix A).

## 3. Results

Figure 1 shows the flow diagram for the selection of the studies. The primary search on the databases returned 4244 studies from which 2938 abstracts were screened, and 294 full texts were assessed for eligibility. After considering the inclusion and exclusion criteria, 33 articles were included in the qualitative synthesis. 

### 3.1. Characteristics of the Selected Studies 

Table 1 describes the main characteristics the selected studies according to the type of exposure (i.e., environmental or occupational). 

#### 3.1.1. Environmental Studies

Twenty environmental studies evaluated the association between PM_2.5_ short-term exposure and changes in FEV_1_ and FVC. Nineteen of these studies measured FEV_1_ while fifteen measured FVC. Most of these studies were performed in North America (**N** = 10), followed by East Asia (**N** = 7) and Europe (**N** = 3). The different contexts of exposure identified were: hourly exposure to particles while commuting [50,51], performing physical exercise [23,25,55,57,58], and associated with different microenvironments [29,30]; daily average exposures varying according to different periods or areas [27,28,49,54,56,60,61,62] and workers’ exposure to ambient particles [53,59]. Although these latter studies were performed in a workers’ population, we have considered that the type of exposure—including levels, composition and sources—was similar to the exposure experienced by individuals from the general environment. Environmental studies were designed predominantly as crossover (**N** = 11) and panel (**N** = 9) studies. Eleven studies estimated exposure by personal/quasi-personal measurements while nine used near/central station measurements. The exposures considered were only related to PM_2.5_ and mean levels exposure ranged between 2 µg/m^3^ and 162 µg/m^3^, while exposure duration ranged between 1 h and 24 h. The mean age of the subjects was 31.2 years old and they were predominantly men (61.4%). Only two of the 20 studies included current smokers in the population.

#### 3.1.2. Occupational Studies 

Thirteen occupational studies investigating associations between exposures to particles during a full work shift and FEV_1_ and FVC changes were included [33,44,45,46,47,48,63,64,65,66,67,68,69]. All thirteen studies measured FEV_1_, while eight also assessed FVC. Most of these occupational studies were carried out in North America (**N** = 4) and the Middle East (**N** = 4), followed by Europe (**N** = 2), Oceania (**N** = 2), and East Asia (**N** = 1). In terms of study design, one was a panel study (**N** = 1), whereas the remaining 12 studies were cross-shift studies (i.e., the health outcome is measured before and after the working shift). Exposure contexts included diesel exhaust; different types of dust such as cotton, wood, and cement; and exposure to particles experienced by dairy workers, firefighters and dental laboratory technicians. All occupational studies assessed exposure by personal or quasi-personal measurements (i.e., measurements were performed close to the worker but not in the breathing zone). The majority of the studies focused on PM_4_ and mean levels of exposure to fine particles ranged between 35 µg/m^3^ and 6760 µg/m^3^, while exposure duration ranged between 6 h and 12 h. The mean age of the subjects was 34.9 years old, and male workers comprised most of the population (86%). The mean percentage of current smokers was 33.8%.

### 3.2. Quality Assessment

The quality assessment of the studies included in the meta-analyses is described in the Appendix A (Appendix A). Based on the quality assessment of exposure, outcome, and cofounding criteria, all selected studies were labeled as good quality and included in the meta-analyses. Specifically, we considered occupational studies to have a higher quality in the exposure assessment criteria because most of them used personal measurements of exposure, compared to many environmental studies that assessed exposure by central stations. On the other hand, environmental studies were qualified as higher quality in the confounding bias criteria (that includes the lack of consideration of important modifiers). The crossover and panel designs of these studies, combined with the inclusion of co-variables in the regression models, allowed the control of important confounders. In occupational studies, however, the influence of important confounders, such as co-exposures, and effect modifiers, such as smoking status, were not considered in some cross-shift studies. No clear differences between environmental and occupational studies were observed for the outcome assessment criteria in relation to the performance of the spirometry maneuvers by a trained technician and according to an official guideline.

### 3.3. Forest Plots and Meta-Analyses

The comparison of environmental and occupational studies was only possible for lung function parameters expressed as absolute changes (mL changes). In total, 14 studies were included in this main set of meta-analyses. Panels (a) and (b) of Figure 2 present forest plots showing separate estimates for FEV_1_ (mL change) of environmental and occupational studies, respectively, whereas panels (a) and (b) of Figure 3 present forest plots for FVC (mL change). Other meta-estimates of associations between FEV_1_-FVC and environmental exposures to PM_2.5_ that could not be compared with occupational studies are presented in Appendix A (Appendix A). 

#### 3.3.1. FEV_1_ (mL Change)

FEV1 and PM2.5 in environmental studies

Panel (a) of Figure 2 shows the forest plot for the six environmental studies that reported associations between PM_2.5_ short-term exposure, with duration between 1 h and 24 h, and FEV_1_ in mL change. Across these studies, the exposure levels varied from 2 µg/m^3^ (cycling indoors; Weichenthal et al. (2011) [25]) and 146.5 µg/m^3^ (average of daily concentrations in 2 cities of China; Hao et al. 2017) [49]). A 10 µg/m^3^ increase in exposure levels was associated with a reduction of 7.63 mL (95% CI: −10.62 to −4.63 mL) in FEV_1_, with no heterogeneity across results given the substantial overlap of confidence intervals (I^2^ = 0%). The 95% prediction interval indicates that estimates of similar future studies would be expected to be between −10.62 mL and −4.63 mL. In the leave-one-out test, the exclusion of Vilcassim [60], the study carrying the higher weight (56%), did not meaningfully affect the meta-estimate: −8.43 mL (95% CI: −12.96 to −3.89; I^2^ = 0%). The exclusion of Hao [49], the study with the highest PM_2.5_ concentration, also did not affect the interpretation of the model: −7.01 mL (95% CI: −10.62 to −3.41; I^2^ = 0%). In addition, Weichenthal et al. (2011) [25] reported 33% of asthmatics in the studied population, and the exclusion of this study did not affect the meta-estimate: −7.61 mL (95% CI: −10.61 to −4.61; I^2^ = 0%). Forest plots grouped by exposure duration, study design, and type of measurement are presented in the Appendix A (Appendix A); estimates based on central sites and daily exposures (24 h) had much smaller confidence intervals. 

FEV1 and PM4 in occupational studies

Panel (b) of Figure 2 shows the forest plot for eight occupational studies that reported associations between short-term exposure to PM_4_ and FEV_1_ in mL change. Across these studies, the exposure levels varied from 270 µg/m^3^ (wood dust exposure; Herbert et al. (1994) [63]) and 2390 µg/m^3^ (cotton dust exposure; Bakirci et al. (2007) [45]). A negative association was observed, but the meta-estimate was lower compared to environmental studies; a 10 µg/m^3^ increase in PM_4_ concentration was associated with a reduction of 0.87 mL (95% CI: −1.36 to −0.37 mL) in FEV_1_ after a work shift. Heterogeneity across results was moderate (I^2^= 54%). The 95% prediction interval indicates that the estimate of similar future studies would be expected to be between −1.85 mL and 0.11 mL. The removal of Bakirci [45], the study with the highest average dust concentration, reduced the heterogeneity of the model but did not affect the interpretation of the estimate: −1.14 mL (95% CI: −1.83 to −0.45; I^2^ = 45%). Barkirci et al. (2007) [45], Bakirci et al. (2006) [46] and Altin et al. (2002) [44] reported a percentage of non-healthy workers of 20%, 14%, and 11.5%, respectively. The exclusion of these studies reduced the heterogeneity of the model but did not affect the interpretation of the estimate: −0.76 mL (95% CI: −1.34 to −0.18; I^2^ = 18%).

#### 3.3.2. FVC (mL Changes)

FVC and PM2.5 in environmental studies

Panel (a) of Figure 3 shows for FVC, expressed as mL changes, five environmental studies that could be pooled. For an increment of 10 µg/m^3^ in PM_2.5_ exposure, the random-effect meta-estimate showed a reduction of 10.0 mL (95% CI: −18.62 to −1.37 mL) in FVC. Although there were substantial differences across primary mean effect estimates, statistical heterogeneity was low (I^2^= 27%) given the wide confidence intervals, particularly for three studies. The 95% prediction interval indicates that the estimate of similar future studies would be expected to be between −22.9 mL and 2.9 mL. The exclusion of Weichenthal [25], which included non-healthy individuals, increased the heterogeneity and caused the confidence interval to include the zero value: −9.59 mL (95% CI: −19.81 to 0.63; I^2^= 45%). Forest plots grouped by exposure duration, study design, and type of measurement are presented in the Appendix A (Appendix A); no clear trend was seen from this grouping.

FVC and PM4 in occupational studies

Panel (b) of Figure 3 presents the forest plot of the occupational studies associating exposures to PM_4_ and mL changes in FVC. Since only three studies were included, pooled estimates were not calculated and only results of the individual studies are presented. All studies reported a reduction in FVC levels after a work shift. Estimates ranged from −12.3 mL to −1.37 mL for a 10 µg/m^3^ increase in PM_4._

### 3.4. Publication Bias

Funnel plots are presented in the Appendix A (Appendix A). In general, there are not enough studies to comprehensively examine publication bias. Visual inspections of the funnel plots revealed no strong indication of publication bias, although it cannot be excluded.

## 4. Discussion

This review is the first to consider associations from occupational and environmental health studies investigating short-term exposure to fine particles on FEV_1_ and FVC in healthy adults. Our analysis shows that exposure to fine particles is associated with reductions in FEV_1_ and FVC among healthy adults in both occupational and environmental exposure settings. For a similar exposure increment (10 µg/m^3^), the associations with fine particles in healthy adults are an order of magnitude greater in environmental studies as compared to occupational studies. Even if PM exposure in occupational settings were from very diverse settings, the estimate for FEV_1_ was relatively consistent considering the varied exposure contexts.

Two hypotheses may explain the 10-fold difference in the magnitude of the occupational and environmental meta-estimates of FEV_1_ for the same exposure increment. Firstly, this difference may reflect the distinct characteristics between occupational and environmental studies, notably in the composition of particles due to the varied sources of ambient versus workplace exposures, size fraction (i.e., PM_2.5_ versus PM_4_), sampling strategy (i.e., personal monitoring versus central station), study design (i.e., cross-shift versus panel studies), exposure duration (i.e., daily versus hourly), study population (i.e., sex and smoking status), and the healthy worker effect [70]. In this regard, almost all environmental studies excluded smokers from the population, while occupational studies included smokers. Given that the association between PM and lung function can differ according to smoking status [71], this factor may also partially explain the difference observed between environmental and occupational studies. In addition, the assessment of co-exposures that are also relevant to lung function effects was not explored by many occupational studies. Another possible explanation is that the variation in the acute response of the airways depends on the condition of the lungs that could be damaged by chronic exposures.

However, although these factors may explain a portion of the observed difference, they may not fully explain the almost 10-fold difference between both meta-estimates. In this regard, another hypothesis may be related to differences in effects according to the range of PM concentrations in occupational (i.e., between 270 µg/m^3^ and 2390 µg/m^3^) and environmental (i.e., between 2 µg/m^3^ and 146.5 µg/m^3^) studies. Indeed, there may be a nonlinear relationship linking PM exposure to lung function, with a steeper slope at lower concentrations (i.e., environmental exposure) that may flatten in the higher ranges, as observed in some mortality studies with ambient fine particles [72,73]. Biologically, this could indicate that high short-term exposure levels—such as observed in occupational studies—could lead to the saturation of cellular and biochemical mechanisms involved in acute lung inflammation and oxidative stress, resulting in a plateau in the exposure-response relationship at these concentrations [11,72].

Despite the low number of studies included in this review, which might limit the external validity, this study is the first to review the effect of short-term exposure to fine particles on FEV_1_ and FVC from occupational studies. Findings from our meta-analysis of environmental studies are in accordance with a recent meta-analysis published that showed a reduction of 7.02 mL (95% CI: −11.75 mL to −2.29 mL) in FEV_1_ after short-term environmental exposures to PM_2.5_ in healthy adults [74]; while we observed a reduction in FEV_1_ of 7.62 mL (95% CI: −10.62 to −4.73 mL) in environmental studies. In contrast to this review of environmental studies, our analysis benefits, in terms of causal inference, of being restricted to studies involving repeated measurements (i.e., panel, crossover and occupational cross-shift but excluding cross-sectional). Studies with repeated measurements enable adequate assessment of the variation in lung function that is attributed to short-term variations in air pollution by accounting for the baseline lung function and controlling for the possible long-term effect of air pollution on lung function. We further improved over this previous review by including studies on FVC, by considering all outcome units (e.g., % change) and reviewing occupational studies, which reinforce the findings that short-term exposure to fine particles leads to decrement in lung function in healthy adults.

Future studies are needed to improve our understanding of the impacts of daily particulate exposure in both occupational and general environments. Notably, the clinical relevance of small daily changes in FEV_1_ in healthy adults remains unclear, although reductions in lung function parameters are suggested as a predictor for cardiopulmonary mortality and morbidity [75]. Furthermore, the minimal clinical important difference (MCID) for clinical trials in patients with COPD is 5% or 100 ml [76], which is a decrease observed in some occupational studies reported here [44,45,46,48]. Other research questions that need to be addressed include how high daily exposure levels may influence the duration and transience of respiratory effects and whether the short-term effects from repeated daily exposures are also linked to the longitudinal decline in lung function and the development of cardiopulmonary morbidities. In this regard, it is suggested that short-term PM exposure may lead to an increased tonus of airway smooth muscles that is typically rapidly antagonized by an increased cellular level of nitric oxide (NO), resulting in transitory airway resistance [77]. This may explain why, for some studies, short-term exposures to particles did not result in significant reductions in lung function.

The impact of PM composition from different sources on lung function may be addressed in future studies by oxidative potential assays and compared across occupational and environmental contexts. Furthermore, panel studies with repeated measurements across different days could also be developed for occupational settings. This type of study design would be fundamental to understand how occupational exposures across different days (i.e., with different lags for effects) affects the duration and transience of lung function reductions in workers. 

Routine and harmonized measurements of occupational exposures to UFPs are necessary to compensate for the general lack of data that prevents establishing exposure matrix and acceptable levels of exposure [35]. To date, occupational exposures to fine particles are not specifically targeted by occupational health regulation [35,78] and our findings strongly suggest that daily levels of exposure in workplaces should be controlled. Practitioners should consider exposure to fine particles as a potential hazard related to respiratory symptoms in their patients.

## 5. Conclusions

This systematic review and meta-analysis show that environmental and occupational short-term exposures to fine particles are associated with reduced FEV_1_ and FVC in healthy adults. A lower meta-estimate was found in occupational studies than environmental studies for a similar exposure increment; however, exposure levels were substantially greater in occupational studies. This may reflect a potentially nonlinear relationship linking PM exposure to certain lung functions parameters, with a steeper slope at lower concentrations. Differences in meta-estimates may also be, in part, due to differences across occupational and environmental study design and methods. Future meta-analyses would benefit from greater standardization of study design and methods, notably in terms of the metric used to express the lung function parameters and the fraction of particles measured.

## Figures and Tables

**Figure 1 ijerph-18-10571-f001:**
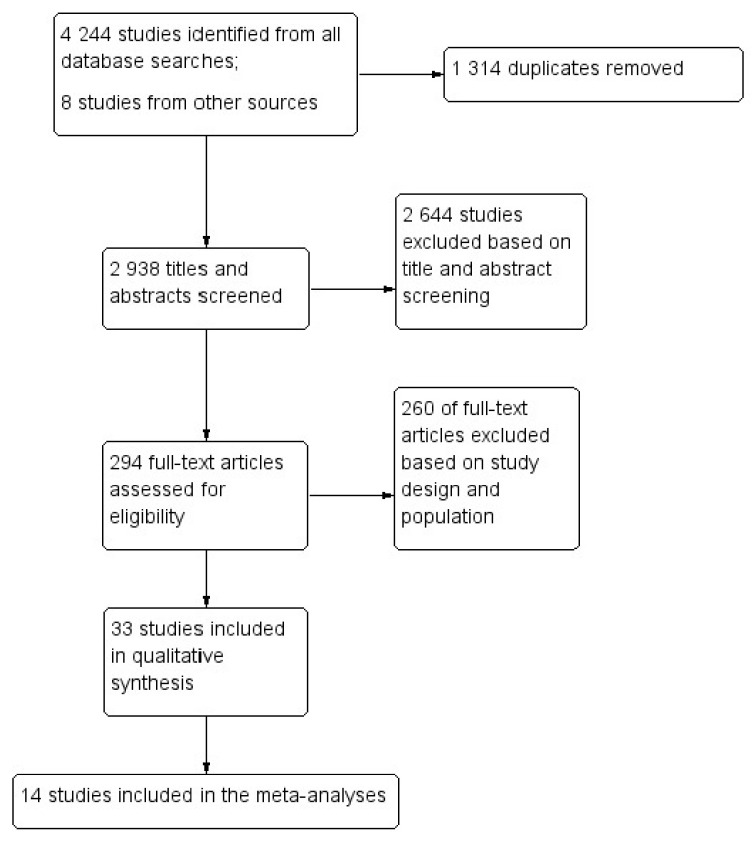
Flow diagram for the selection of studies.

**Figure 2 ijerph-18-10571-f002:**
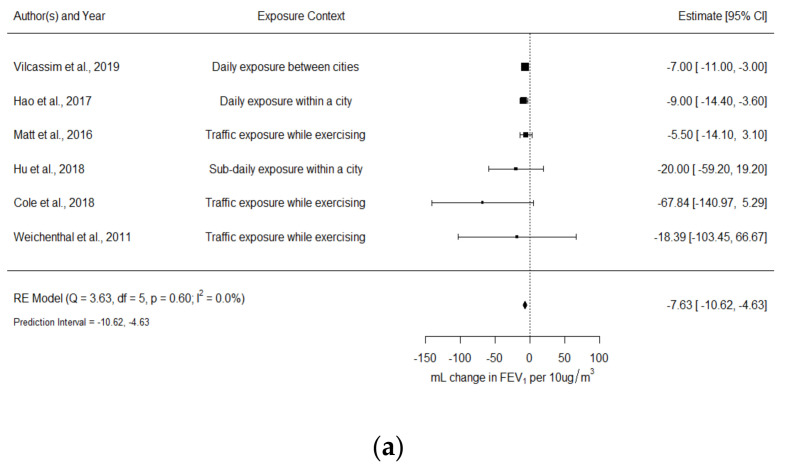
Forest plots of the association between (**a**) environmental PM_2.5_ and FEV_1_ (mL change); (**b**) occupational PM_4_ and FEV_1_ (mL changes). Random-effect meta-estimate of association is indicated by vertical point of diamond and 95% CI is represented by horizontal point. Squares represent individual effect size of primary studies and the bars the 95% CI; size of squares is proportional to weight in calculating random-effect summary estimates. Pooled effect sizes were estimated per 10 µg/m^3^ increase in exposure level. Daily exposure is defined as a 24 h exposure duration, while sub-daily is defined as an exposure duration <24 h.

**Figure 3 ijerph-18-10571-f003:**
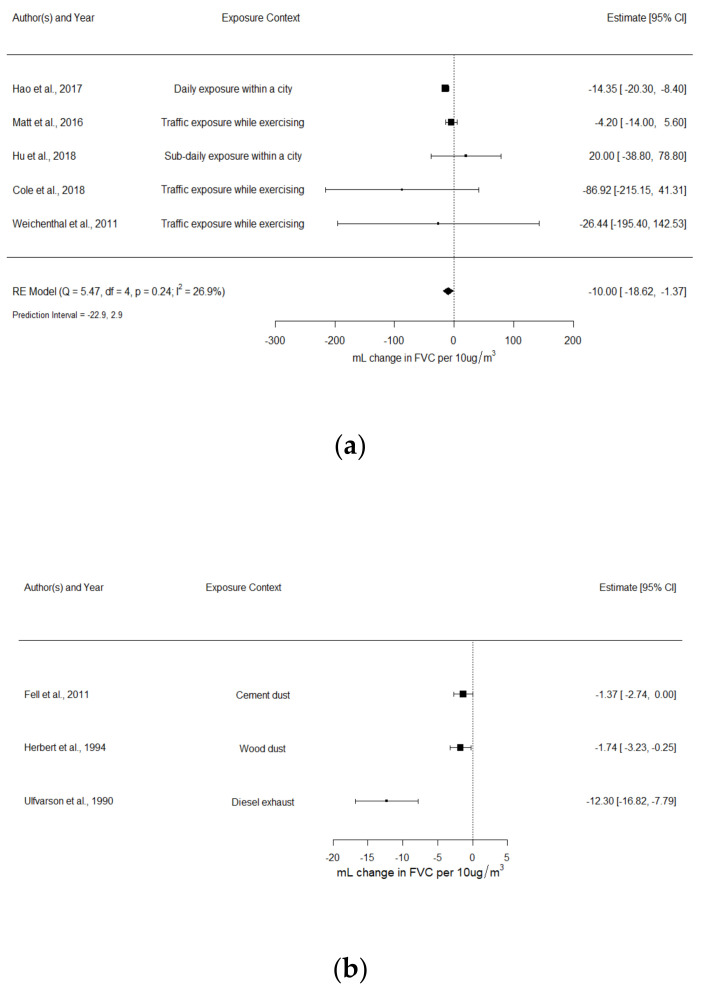
(**a**) Environmental PM_2.5_ and FVC (mL changes); (**b**) occupational PM_4_ and FVC (mL changes). Random-effect meta-estimate of association is indicated by vertical point of diamond and 95% CI is represented by horizontal point. Squares represent individual effect size of primary studies and the bars the 95% CI; size of squares is proportional to weight in calculating random-effect summary estimates. Pooled effect sizes were estimated per 10 µg/m^3^ increase in exposure level. Daily exposure is defined as a 24 h exposure duration, while sub-daily is defined as an exposure duration <24 h.

**Table 1 ijerph-18-10571-t001:** Characteristics of the studies included in the systematic review and meta-analyses.

Author and Year	Location	Design	N	Men, %	Age, years	Smokers, %	Exposure Context	Measurement and Exposure Duration	Pollutants and Mean Concentration
Environmental Studies									
Baccarelli et al. (2014) [53]	China	Panel	120	66%	33	39.1	Traffic-related PM2.5 exposure in truck drivers and office workers	Personal: 8 h	PM_2.5_: 127 µg/m^3^ (drivers) and 94 µg/m^3^ (office workers)
Cakmak et al. (2014) [54]	Canada	Crossover	61	46%	24	0	Near steel plant and college campus	Near Station: 24 h	PM_2.5_: 12.8 µg/m^3^ (plant) and 11.5 µg/m^3^ (campus)
* Cole et al. (2018) [23]	Canada	Crossover	38	74%	29	0	Cycling in downtown (D) and residential (R) areas	Quasi-personal: 1 h	PM_2.5_: 6 µg/m^3^ (D) and 4.7 µg/m^3^ (R)
Dales et al. (2013) [28]	Canada	Crossover	61	75%	24	0	Exposure near steel plant and college campus	Near Station: 24 h	PM_2.5_: 12.8 µg/m^3^ (plant) and 11.5 µg/m^3^ (campus)
^†^ Girardot et al. (2006) [55]	USA	Panel	354	43%	43	0	Exposure while hiking in a mountain	Near Station: 5 h	PM_2.5_: 15 µg/m^3^
* Hao et al. (2017) [49]	China	Panel	42	62%	55	0	Daily exposures to particles	Personal: 24 h	PM_2.5_: 146.5 µg/m^3^
* Hu et al. (2018) [56]	China	Panel	28	43%	20.6	0	Same day exposure to particles	Personal: 8 h	PM_2.5_: 65.1 µg/m^3^
^†^ Huang et al. (2016) [29]	China	Crossover	40	42%	24	0	Exposure in a transport hub and park	Personal: 2 h	PM_2.5_: 162 µg/m^3^ (transport hub) and 53 µg/m^3^ (park)
Jarjour et al. (2013) [57]	USA	Crossover	73	73%	32	0	Cycling on low traffic (LT) and high traffic (HT) routes	Personal: 2 h	PM_2.5_: 45 µg/m^3^ (LT) and 44 µg/m^3^ (HT)
Kubesch et al. (2015) [58]	Spain	Crossover	28	46%	34	0	Exposure to high and low TRAP in combination with physical exercise	Quasi-Personal: 2 h	PM_2.5_: 30 µg/m^3^ (LT) and 80.1 µg/m^3^ (HT)
Liu et al. (2018) [27]	Taiwan	Panel	100	50%	46	0	Daily exposure to particles	Central Station: 24 h	PM_2.5_: 25.6 µg/m^3^
* Matt et al. (2016) [24]	Spain	Crossover	30	50%	36	0	Exposure in high traffic (HT) and low traffic (LT) roads while performing physical activity	Near Station: 2 h	PM_2.5_: 39 µg/m^3^ (LT) and 82 µg/m^3^ (HT)
Mirabelli et al. (2015) [50]	USA	Crossover	21	62%	35	0	Exposure while commuting	Quasi-Personal: 2 h	PM_2.5_: 28.8 µg/m^3^
^†^ Mirowsky et al. (2015) [30]	USA	Crossover	23	48%	25	0	Walking near traffic routes	Quasi-Personal: 2 h	PM_2.5_: 20 µg/m^3^; PM_10_: 26 µg/m3
^†^ Thaller et al. (2008) [59]	USA	Panel	142	79%	19	27	Beach guards exposed to ambient PM_2.5_	Central Station: 8 h	PM_2.5_: 10.7 µg/m^3^
* Vilcassim et al. (2019) [60]	USA	Panel	34	32%	27	0	Exposure in different cities while travelling by plane	Central Station: 24 h	PM_2.5_: From 8.7 µg/m^3^ (New York) to 105 µg/m^3^ (East Asia)
* Weichenthal et al. (2011) [25]	Canada	Crossover	42	67%	35	0	Cycling indoors, low traffic (LT) and high traffic routes (HT)	Quasi-Personal: 1 h	PM_2.5_: 2 µg/m^3^ (Indoor), 8.1 µg/m^3^ (LT) and 44 µg/m^3^ (HT)
^†^ Wu et al. (2013 a) [61]	China	Panel	40	100%	20	0	Exposure in suburban and urban areas	Central Station: 24 h	PM_2.5_: 75.2 µg/m^3^ (Suburban), 56.6 µg/m^3^ (Urban 1) and 48.8 µg/m^3^ (Urban 2)
^†^ Wu et al. (2013 b) [62]	China	Panel	21	100%	20	0	Exposure in suburban and urban areas	Central Station: 24 h	PM_2.5_: 75.2 µg/m^3^ (Suburban), 56.6 µg/m^3^ (Urban 1) and 48.8 µg/m^3^ (Urban 2)
^†^ Zuurbier et al. (2011) [51]	Netherlands	Crossover	34	70%	42	0	Commuting by bus, car, and by bike	Quasi-Personal: 2 h	PM_2.5_: 58 µg/m^3^ (vehicles) and 65.2 µg/m^3^ (bike)
**Occupational Studies**									
* Altin et al. (2002) [44]	Turkey	Cross-shift	223	78%	27	67	Occupational exposure to cotton dust	Personal: 8 h	PM_4_: 413 µg/m^3^
* Bakirci et al. (2006) [46]	Turkey	Cross-shift	66	100%	NA	79	Occupational exposure to cotton dust	Quasi-Personal: 8 h	PM_4_: 1050 µg/m^3^ (delinting), 1870 µg/m^3^ (hulling) and 610 µg/m^3^ (baling)
* Bakirci et al. (2007) [45]	Turkey	Cross-shift	157	20%	52	31.2	Occupational exposure to cotton dust	Personal: 8 h	PM_4_: 2390 µg/m^3^
* Fell et al. (2011) [47]	Norway	Cross-shift	70	92%	41	41	Occupational exposure to cement dust	Personal: 8 h	PM_4_: 300 µg/m^3^
* Gaughan et al. (2014) [33]	USA	Cross-shift	17	94%	26	0	Firefighters exposed to particles	Personal: 12 h	PM_4_: 490 µg/m^3^
* Herbert et al. (1994) [63]	Canada	Cross-shift	99	NA	35	27.9	Occupational exposure to wood dust	Quasi-Personal: 6 h	PM_4_: 270 µg/m^3^
Hu et al. (2006) [64]	Taiwan	Panel	45	66%	30	31.3	Exposure in dental laboratories	Personal: 8 h	PM_2.5_: 107 µg/m^3^
Mandryk et al. (1999) [65]	Australia	Cross-shift	198	100%	37	33	Occupational exposure to wood dust	Personal: 8 h	PM_4_: 2170 µg/m^3^ (sawmill) and 1700 µg/m^3^ (joinery)
Mandryk et al. (2000) [66]	Australia	Cross-shift	127	100%	36	47.1	Occupational exposure to wood dust	Personal: 8 h	PM_4_: 2260 µg/m^3^ (green mill) and 1460 µg/m^3^ (dry mill)
Mitchell et al. (2015) [67]	USA	Cross-shift	205	100%	34	24.4	Dairy workers exposed to particles	Personal: 9.2 h	PM_2.5_: 35 µg/m^3^ (Workers) and 19.6 µg/m^3^ (Controls)
Neghab et al. (2018) [68]	Iran	Cross-shift	200	100%	37	41	Occupational exposure to wood dust	Personal: 8 h	PM_4_: 6760 µg/m^3^
* Slaughter et al. (2004) [69]	USA	Cross-shift	65	80%	29	16.9	Firefighters exposed to particles	Personal: 8 h	PM_4_: 880 µg/m^3^
* Ulfvarson and Alexandersson (1990) [48]	Sweden	Cross-shift	24	100%	35	0	Exposure to diesel exhaust	Quasi-Personal: 8 h	PM_4_: 240 µg/m^3^

Abbreviations: N: number of subjects; h: hours; NA: Not available; PM2.5: Particulate matter with median diameter of less than 2.5 µm; PM4: Particulate matter with median diameter of 4 µm. * Studies included in the main set of meta-analyses. † Studies included in the meta-analyses of the Appendix A. Results of the studies with no symbols are presented in Appendix A.

## Data Availability

The datasets used and/or analyzed during the current study are available from the corresponding author on reasonable request.

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
