# Peer review of "Environmental and Occupational Short-Term Exposure to Airborne Particles and FEV_1_ and FVC in Healthy Adults: A Systematic Review and Meta-Analysis"

_ijerph, 2021, doi:10.3390/ijerph182010571_

Round 1
Reviewer 1 Report
This Study estimated the associations between short term exposures to fine pmanuscripts (PM2.5 and PM4) and lung function (FEV1 and FVC) in healthy adults. The manuscript is very interesting and has a certain meaning. However, before publication, major revisions are required. The following is my personal suggestions:
- It is recommended to change the expression of the topic to make it more accurately summarize the research content.
- 2. Line 39-40. Is there a lack of such research or a lack of review studies of such research?
- 3. The introduction part is relatively shallow. It is recommended to add more content about research progress in addition to the research background.
3Line60-61. Why only choose these two databases to search for documents?
- Combined with the content mentioned in the Introduction, can the two lung function parameters of the current search terms completely represent lung function? If not, then the title and part of the text are not rigorous enough, and the words need to be corrected. Therefore, the statement that there is no relevant research currently written in the Introduction is not rigorous enough, and it is recommended to further review the existing research. It is recommended to make the title and the wording in the text more rigorous.
- It is recommended to write the Results and Discussion sections together, and combine the Forrest plots of the two studies, so that the author can analyze the results and show the comparison results for readers.
- Line229-230, why do the selected studies have sufficient quality? It is recommended to supplement the basis of this view.
Author Response
Reviewer 1
This Study estimated the associations between short term exposures to fine pm (PM2.5 and PM4) and lung function (FEV1 and FVC) in healthy adults. The manuscript is very interesting and has a certain meaning. However, before publication, major revisions are required. The following is my personal suggestions:
1. It is recommended to change the expression of the topic to make it more accurately summarize the research content.
From our understanding, the reviewer would like the topic of our research to be more specific about the parameters studied, FEV1 and FVC, which do not exclusively characterize lung function. We have made various changes in the title and in the text to replace the term lung function with FEV1 and FVC specifically.
2. Line 39-40. Is there a lack of such research or a lack of review studies of such research?
There is a lack of review studies of such research. We have added the word “systematically” to the sentence for clarification.
3. The introduction part is relatively shallow. It is recommended to add more content about research progress in addition to the research background.
We have revised the introduction to add content on the burden of disease, pollution and more specifically particulate matter, health effects associated with general, short-term, and long-term PM exposure.
4. Line60-61. Why only choose these two databases to search for documents?
The majority of studies retained for the analysis were retrieved from Pubmed and Web of Science was relevant for occupational studies in small journals not captured by the Pubmed. These two databases are often complementary and have been chosen by others to perform systematic reviews (https://pubmed.ncbi.nlm.nih.gov/28763933/).
It is not unusual for reviews to focus on two databases (https://www.sciencedirect.com/science/article/pii/S0160412020319292?via%3Dihub ; https://www.tandfonline.com/doi/abs/10.1080/15412555.2016.1216956) and recent published systematic reviews on air pollution have only used Pubmed (e.g. https://pubmed.ncbi.nlm.nih.gov/27565881/ )
5. Combined with the content mentioned in the Introduction, can the two lung function parameters of the current search terms completely represent lung function? If not, then the title and part of the text are not rigorous enough, and the words need to be corrected. Therefore, the statement that there is no relevant research currently written in the Introduction is not rigorous enough, and it is recommended to further review the existing research. It is recommended to make the title and the wording in the text more rigorous.
We made the wording more accurate and now only refer to FEV and FVC specifically. We have modified the introduction to provide more context.
6. It is recommended to write the Results and Discussion sections together, and combine the Forrest plots of the two studies so that the author can analyze the results and show the comparison results for readers.
We have presented the results and discussion separately, according to journal guidelines.
The forest plots have been combined for comparison.
7. Line229-230, why do the selected studies have sufficient quality? It is recommended to supplement the basis of this view.
This statement is based on the results of the quality assessment of the exposure, outcome and cofounding criteria. We have changed the sentence to read: “Based on the quality assessment of exposure, outcome and cofounding criteria, all selected studies were labelled as good quality and included in the meta-analyses.”
Reviewer 2 Report
The authors reviewed in the manuscript, nice and carefully, the impact on lung function specific tests (in healthy adults) of environmental and occupational
short-term exposures to fine particles. My questions/comments/suggestions are:
1 - the authors report some covariables in table 1 (mean age, sample size, Male % , Smokers %) that might also explain some of the heterogeneity found. Does this covariables (or others like longitude)
were meta-regressed against the outcomes ? It would be interesting to know about it.
2 - Table S4 reports the risk of bias results but it is not easy to read. Consider ilustrate the risk of bias results (also) as traffic light plots for the studies selected for quantitative analysis.
3 - In exclusion criteria: (only) size fractions other than PM2.5 and PM4 were measured;
4 - prediction intervals are sensitive to the number of studies in the meta-analysis and change according to the estimating method... what was the estimating method used here?
5 - Since the initial goal was to compare between environmental and occupational settings, meta-analysis results would be better compared
through a sub-group forest plot for settings, for each lung test / particle size combination, instead of standard forest plots.
Furthermore, the sub-group model between settings heterogeneity would be a elegant way to evaluate differences between the two settings,
that could also be further evaluated through z-testing between the two settings pooled results. That would give statistical
support to the conclusion ("A lower meta-estimate was found in occupational studies than environmental studies for a similar exposure increment")
6 - Lower sample sized studies seem to be overexpressing reduction in lung function as measured by FEV1 global meta-analysis results (in funnel plots for FEV1). Might be
interesting to discuss this partial sign of publication bias or look for other causes supporting this trend.
7 - There are some format/style inconsistency across figures that need to be solved.
Author Response
1 - the authors report some covariables in table 1 (mean age, sample size, Male % , Smokers %) that might also explain some of the heterogeneity found. Does this covariables (or others like longitude) were meta-regressed against the outcomes ? It would be interesting to know about it.
We added the following sentence to the method: “A meta-regression of the covariables against the outcome was not performed as some parameters (such as age) were very similar between studies, or even missing (smoking status), and sample size was too small.”
2 - Table S4 reports the risk of bias results but it is not easy to read. Consider ilustrate the risk of bias results (also) as traffic light plots for the studies selected for quantitative analysis.
We did the risk of bias assessment to identify some studies that may have been of poor quality but we didn’t want to emphasize it. Because we didn’t identify studies of low quality, we do not believe it is worth illustrating in other forms. We have however added colours to the existing table to facilitate the reading.
3 - In exclusion criteria: (only) size fractions other than PM2.5 and PM4 were measured;
The term “only” was added.
4 - prediction intervals are sensitive to the number of studies in the meta-analysis and change according to the estimating method... what was the estimating method used here?
Meta-analysis was done with random-effect models (mentioned in line 161) based on the DerSimonian and Laird method (mentioned in lines 184-185) using the metafor package in R (mentioned in line 186).
5 - Since the initial goal was to compare between environmental and occupational settings, meta-analysis results would be better compared through a sub-group forest plot for settings, for each lung test / particle size combination, instead of standard forest plots.
We have combined the plots as suggested by the reviewer to facilitate the comparison.
Furthermore, the sub-group model between settings heterogeneity would be a elegant way to evaluate differences between the two settings, that could also be further evaluated through z-testing between the two settings pooled results. That would give statistical support to the conclusion ("A lower meta-estimate was found in occupational studies than environmental studies for a similar exposure increment")
We acknowledge the relevance of such testing, however, the data we have does not allow us to make such comparison for the two settings. Indeed, due to the small number of studies, we were not able to calculate pooled estimates for FVC and PM4 in occupational studies. However, we performed a Wald-type test for FEV1, which showed that the mean meta-estimate from occupational studies was significantly smaller than that from environmental studies (p=<0.001).
6 - Lower sample sized studies seem to be overexpressing reduction in lung function as measured by FEV1 global meta-analysis results (in funnel plots for FEV1). Might be interesting to discuss this partial sign of publication bias or look for other causes supporting this trend.
Studies that show a greater reduction in FEV1 do not necessarily have the smallest samples among the studies selected for meta-analysis. The study from Cole et al. which shows the largest reduction in FEV1 in relation to exposure to PM2.5 had a sample of N = 38, while a smaller reduction in FEV1 was obtained in the study by Hu et al. with N = 28. It is therefore not certain that the lower sample-sized studies are associated with over reduction in lung function.
We have however modified the paragraph of the Publication bias in the results section as follow: Visual inspections of the funnel plots revealed no strong indication of publication bias, although it cannot be excluded. ”
7 - There are some format/style inconsistency across figures that need to be solved.
We made the corrections.
Round 2
Reviewer 1 Report
The author has made some changes to the article. It is recommended that the author can carefully modify and improve the quality of the in accordance with the review comments of the reviewers, including the format and content.
Author Response
The author has made some changes to the article. It is recommended that the author can carefully modify and improve the quality of the in accordance with the review comments of the reviewers, including the format and content.
We thank the reviewer for his/her contribution to the review of this manuscript.
We would like to respond to the reviewer's request, but we are confused by the last comment and do not understand the scope of it. In our first review of the manuscript, we took each comment into consideration and made changes accordingly.
We replaced the terms pulmonary function to refer only to FEV1 and FVC, which are two indices used for the measurement of ventilatory function (Pierce, R. (2005)) and the most commonly analyzed to describe lung function (Knuiman, M. W., 1991 ; Chu, L. M., 2020; Ivanova, O., 2020; Ratanachina, J., 2020); we clarified that there was a lack of systematic review of the literature on the effects of particulate exposure in healthy individuals; we added content to the introduction to make it more comprehensive; we combined the forest plots and we specified that the studies were judged to be of good quality because of the risk bias assessment.
Regarding the format, we have directly used the template provided by the journal, although the guidelines for authors indicate that the journal accepts free format submissions as well.
We would like the Editor to let us know more clearly what else needs to be done for the manuscript to meet expectations.
References
Chu, L. M., Karunanayake, C. P., Dosman, J. A., & Pahwa, P. (2020). Lung Function Decline in Farm and Nonfarm Rural Residents of Saskatchewan. Journal of occupational and environmental medicine, 62(6), e250-e259.
Ivanova, O., Khosa, C., Bakuli, A., Bhatt, N., Massango, I., Jani, I., ... & Rachow, A. (2020). Lung Function Testing and Prediction Equations in Adult Population from Maputo, Mozambique. International journal of environmental research and public health, 17(12), 4535.
Knuiman, M. W., James, A. L., Divitini, M. L., Ryan, G., Bartholomew, H. C., & Musk, A. W. (1999). Lung function, respiratory symptoms, and mortality: results from the Busselton Health Study. Annals of epidemiology, 9(5), 297-306.
Pierce, R. (2005). "Spirometry: an essential clinical measurement." Australian family physician 34(7).
Ratanachina, J., De Matteis, S., Cullinan, P., & Burney, P. (2020). Pesticide exposure and lung function: a systematic review and meta-analysis. Occupational Medicine, 70(1), 14-23.